# Numerical and Experimental Investigation of Slope Deformation under Stepped Excavation Equipped with Fiber Optic Sensors

**Jia Wang** [1], **Wenwen Dong** [1], **Wenzhao Yu** [1], **Chengcheng Zhang** [1] **and Honghu Zhu** [1,2,*]

[1] School of Earth Sciences and Engineering, Nanjing University, Nanjing 210023, China; jiawang@smail.nju.edu.cn (J.W.); dongwenwen@smail.nju.edu.cn (W.D.); yuwenzhao@smail.nju.edu.cn (W.Y.); zhang@nju.edu.cn (C.Z.)

[2] Institute of Earth Exploration and Sensing, Nanjing University, Nanjing 210023, China

\* Correspondence: zhh@nju.edu.cn

**Abstract:** The real-time evaluation of slope stability is a crucial technical issue in foundation excavation and slope construction. However, conventional monitoring methods often fall short of achieving real-time and accurate measurements, which poses challenges to the timely assessment of slope stability. To address this problem, laboratory tests and numerical simulations were jointly used to evaluate slope stability in this paper. In numerical simulations, the finite element method (FEM) results clearly illustrate the distribution and evolution of internal strain during slope excavation, and the limit equilibrium method (LEM) calculates changes in the safety factor. In laboratory tests, the fiber Bragg grating (FBG) sensing technology was employed to monitor the internal strain of the slope in real time. The distribution characteristics of the slope internal strain field under the condition of stepped excavation were analyzed, and the feasibility of strain-based evaluation of slope stability was discussed. The measurements with FBG sensing technology agree well with the numerical simulation results, indicating that FBG can effectively monitor soil strain information. Of great significance is that the maximum horizontal strain of the slope is closely related to the safety factor and can be used to evaluate slope stability. Notably, the horizontal soil strain of the slope provides insight into both the formation and evolution of the critical sliding surface during excavation. The combination of numerical simulation and intelligent monitoring technology based on FBG proposed in this paper provides a reference for capturing strain information inside the slope and realizing real-time assessment and critical warning of slope stability.

**Keywords:** fiber bragg grating (FBG); slope stability; geotechnical monitoring; model test; numerical simulation

## 1. Introduction

Slope stability evaluation is a basic problem in geotechnical engineering, particularly in the context of excavation-induced slope failures in urban construction and mining industries. In the past decades, various slope stability analysis methods and techniques have been developed [1]. However, due to the complex nature of slope evolution processes, it remains challenging to understand the deformation and failure mechanism of slopes comprehensively and systematically. Excavation is a common cause of slope failures in urban construction and mining [2–4], and it is difficult to evaluate slope stability and predict the field performance in real time due to the stress state adjustment within the slope during excavation stages and external stress condition changes, which can affect slope deformation and stability [5,6]. Therefore, monitoring the slope's internal stress state in real-time using geotechnical instrumentation is crucial to make timely assessments of the slope stability and analyzing the movement mechanisms, enabling potential failure warnings for the slope.

Geotechnical instruments have been developed and applied to monitor slope behavior since the 1970s [7]. Traditional monitoring techniques, such as inclinometers, tiltmeters, and extensometers, are easy to operate and have been widely used in geotechnical monitoring, but they cannot achieve real-time measurement. Furthermore, many instruments based on resistance, vibration lines, or inductive sensors have issues with zero drift because their readings are affected by electrical noise in the environment. In addition, these electrical transducers generally have poor moisture resistance and can hardly achieve long-distance measurement [8,9].

In the past decades, fiber optic sensing technology has been increasingly used to monitor the internal strain and deformation of slopes, proving to be a feasible and effective solution. Compared with other sensors, optical fiber sensors have the advantages of small size, high precision, and immunity to electromagnetic interference. They can also establish a fully or quasi-distributed sensing network combining wavelength division multiplexing (WDM) and time division multiplexing (TDM) techniques to achieve real-time measurement and data transmission [10–12]. Compared to field methods, physical modeling has the advantages of shorter-term testing, lower cost, and more intuitive outcomes [13]. In recent years, several laboratory studies have applied fiber optic sensing technology to slope monitoring. For example, to demonstrate the efficacy of fiber optic monitoring, Zhu et al. [14], Song et al. [12], and Sun et al. [15] investigated changes in internal strains in slopes using fiber optic monitoring technology during local loading, slope-cutting, and geogrid reinforcement, respectively. The results of these studies confirmed the accuracy of fiber optic monitoring technology for measuring internal strains in soil. Since the fiber optic cable can only measure the axial strain distributions, it becomes imperative to establish an analytical model that can effectively characterize the relationship between the strain on the optical fibers and the deformation of the soil. Wu et al. [16] proposed a strain integration method to convert strain measurements into shear displacements. Wang et al. [17] proposed a new analytical method for calculating the shear displacement of critical interfaces in landslides, and they demonstrated its feasibility through laboratory shear tests. Xu et al. [18] affixed FBG onto a rubber strip and established the correlation between the bending angle, the strain of the rubber strip, and the surface deformation of the slope model. To monitor the anchor force of the slope anchor system, a new method was proposed to calculate the slope strain and anchor force considering temperature changes based on optical frequency domain reflectometry (OFDR) technology [19] and the relationship between the strain and the anchoring force of the slope anchoring system was successfully established. While these laboratory experiments have shown that the fiber optic sensing monitoring technology can measure the internal strain of the soil and sense soil deformation [20,21], there have been relatively few attempts to monitor slope strain and its relationship with slope stability during slope excavation, posing a pressing need to address the evolution of slope stability in such scenarios.

Numerous fiber optic monitoring techniques have been developed to monitor the deformation of geotechnical structures. These technologies include fiber Bragg grating (FBG), Brillouin optical time domain reflectometry (BOTDR), Brillouin optical time domain analysis (BOTDA), etc. Among these technologies, the quasi-distributed FBG is one of the most widely used techniques. FBG sensors can be connected in series or multiplexed to achieve high-resolution measurements, excellent corrosion resistance, and long-term durability. Additionally, FBG sensors can be used for multiparameter measurements and in sensor networks [9,22,23]. Due to the characteristics of FBG sensors, multiple sensors can be connected in series to measure strains at various locations during slope excavation. By combining these measurements with the results of numerical simulation, the relationship between slope stability and internal strains during excavation is expected to be explored.

In this paper, the capability of quasi-distributed fiber optic sensing technology in slope stability monitoring under slope excavation conditions is investigated through physical model tests and numerical simulation methods. The fiber Bragg grating (FBG) sensor was buried in the soil mass to measure the horizontal strain distribution of the slope model

under stepped excavation. Combined with the finite element method (FEM) and limit equilibrium method (LEM), the relationship between the slope safety factor and the strain parameter is determined, and the feasibility of strain-based evaluation of slope stability is discussed. The feasibility of FBG sensing technology in slope stability evaluation and landslide warning is demonstrated.

## 2. Materials and Methods

### 2.1. Basics of FBG

2.1.1. Principle of the FBG Sensing Technology

The central wavelength of the reflected light of fiber grating depends on the fringe spacing of the grating, and the fringe spacing of the grating depends on the two factors of strain and temperature [24–26]. To measure the actual strain of an object accurately, the reading of fiber optic grating is generally needed for temperature compensation.

Figure 1 shows the working principle of an FBG. According to Bragg's law, when a broadband source of light is injected into the fiber, the FBG reflects a narrow spectral part of light at a certain wavelength, which is dependent on the grating period and the reflective index of the fiber [9,27]. The Bragg wavelength can be calculated by:

$$\lambda_B = 2n_{eff}\Lambda, \tag{1}$$

where $\lambda_B$, $n_{eff,}$ and $\Lambda$ are the reflected Bragg wavelength, effective refractive index, and the grating period, respectively. Considering a standard single-mode silica fiber, $\lambda_B$ changes linearly with the applied temperature and strain. Therefore, the FBG can be used as a temperate or strain sensor considering the following relationship [28].

$$\frac{\Delta\lambda_B}{\lambda_B} = (1 - p^{eff})\Delta\varepsilon + (\alpha + \xi)\Delta T, \tag{2}$$

where $\lambda_B$ is the original Bragg wavelength under strain-free conditions at 0 °C. $\Delta\lambda_B$ is the change in the Bragg wavelength, $p^{eff}$ is the photo-elastic parameter, and $\alpha$ and $\xi$ are the thermal expansion and thermo-optic coefficients, respectively. $\Delta\varepsilon$ and $\Delta T$ are the strain and temperature variations, respectively.

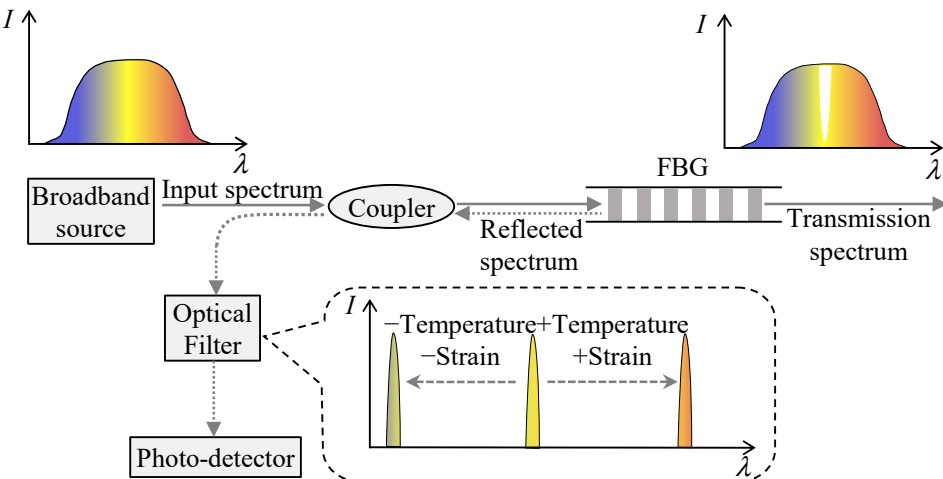

**Figure 1.** Principle of the FBG sensing technique.

2.1.2. Temperature Compensation

Based on the principle of FBG sensing technology, any alterations in strain and temperature are directly reflected in the period of the index modulation and the effective index of refraction of FBG. Variations in environmental temperature can impact the Bragg wavelength, which in turn affects the measured strain data. Therefore, compensating for

temperature is an essential task for any FBG monitoring system. Two commonly used methods for temperature compensation in this area are direct and indirect methods [9].

Standard FBG sensors are unable to simultaneously measure temperature and strain. To compensate for temperature changes, the direct method involves utilizing a resistance thermometer, ROTDR fiber, or similar devices to capture temperature changes. To compensate for temperature changes when measuring strain using FBG sensors, the following equation can be utilized:

$$\Delta \varepsilon = \frac{1}{c_\varepsilon} \left( \frac{\Delta \lambda_B}{\lambda_B} - c_T \Delta T \right), \tag{3}$$

where $c_\varepsilon$ and $c_T$ are the calibration coefficients of strain and temperature, respectively.

The indirect compensation method involves compensating for strain using additional FBG sensors placed in the same sensing area. While the additional sensor is not impacted by mechanical strain-induced soil or rock deformation, it is affected by temperature changes, which are used to achieve temperature compensation by discriminating the wavelength offset. The resulting strain measurements are then corrected as follows:

$$\Delta \varepsilon = \frac{\Delta \lambda_B^\varepsilon}{c_\varepsilon \lambda_B} = \frac{\Delta \lambda_B - \Delta \lambda_B^T}{c_\varepsilon \lambda_B}, \tag{4}$$

where $\Delta \lambda_B^T$ is the shift in wavelength due to temperature change.

### 2.2. Numerical Simulations

For this study, the limit equilibrium method (LEM) and finite element method (FEM) were used to analyze the strain distribution and slope stability of a homogeneous soil slope during excavation using SLOPE/W and PLAXIS$^{2D}$ software, respectively. The slope model had an initial height of 1 m, an angle of 90°, and a slope angle of 76° after excavation. The soil parameters used are listed in Table 1.

**Table 1.** Soil parameters used in the numerical simulation.

| Unit Weight $\gamma$ (kN/m$^3$) | Young's Modulus $E$ (MPa) | | | Shear Strength | | |
|---|---|---|---|---|---|---|
| | $E_{50}$ | $E_{\mathbf{oed}}$ | $E_{\mathbf{ur}}$ | Cohesion $c$ (kPa) | Dilatancy Angle $\psi$ (°) | Friction Angle $\varphi$ (°) |
| 13.2 | 40 | 40 | 120 | 1.6 | 15.2 | 30.4 |

In the LEM analysis models, two limit equilibrium methods (LEM), the Swedish slice method and the Bishop simplified method, were employed to analyze the stability of a slope using the commercial slope stability analysis software SLOPE/W [29,30]. The results of the analyses were used to determine the most critical failure surface and the corresponding safety factor.

In the FEM analysis model, the two-dimensional (2D) finite element model was also used to investigate the stability of the slope. The soil parameters are consistent with those used in the LEM analysis and are listed in Table 1. The boundary conditions of the model are as follows: the bottom was the standard fixed boundary, and the level of the two sides was restrained. The Harding-soil model was used to describe the soil behavior, and the load type was set to the fractional steps construction. The FEM analysis transformed the continuous slope body into a discrete unit, and the slope profile was divided into a finite element unit.

### 2.3. Model Material and Instrumentation

To verify the effectiveness of FBG sensing technology in monitoring slope excavation and evaluating the stability and strain development pattern of slopes under excavation conditions, a physical model test at 1 g condition was performed in the geotechnical

physical modeling laboratory. As shown in Figure 2, the homogeneous and isotropic slope model used in the experiment had dimensions of 1 m (length), 0.5 m (width), and 1 m (height). The model box used in this experiment had an aluminum alloy frame and a rectangle. To facilitate the observation of the slope soil deformation during the test, a 10 mm thick transparent tempered and ribbed glass is installed on the long side, while the short side and bottom are fitted with a single 10 mm thick aluminum alloy plate each.

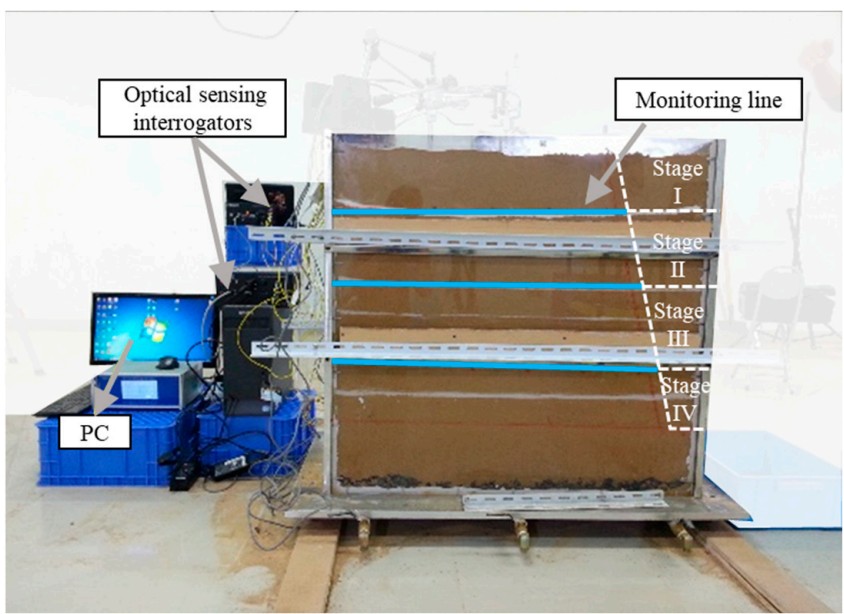

**Figure 2.** Instrumentation of the 2D slope model test.

The soil used in the model tests was sandy-graded sand (SW) collected at a construction site in Nanjing, China. This soil has an initial water content of 3.52%, cohesion of 1 kPa, and internal friction angle of 30.44°, as determined by direct shear tests. The grain size distribution curve showed an average grain size of 0.35 mm with $C_u$ and $C_c$ values of 2.059 and 1.413, respectively.

To simulate plane strain conditions in the experiment, a thin layer of lubricating oil was applied to the inside of the model box. The bottom of the model box was covered with a 10 cm thick layer of gravel, and the soil samples were evenly compacted as the filling thickness reached 3–5 cm. The process was repeated until the full height of the slope model was reached. As shown in Figure 3, the slope model has a height of 0.8 m, a base height of 0.2 m, and a slope angle of 76°.

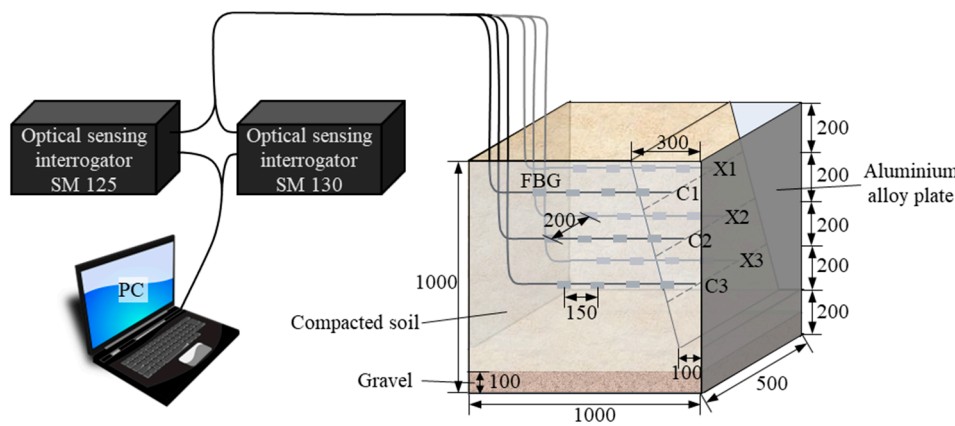

**Figure 3.** Layout of the FBG strain sensing arrays in the slope model (unit: mm).

For the experiment, the FBG sensor demodulation system used two sensing interrogators, SM130 and SM125, produced by Micron Optics Inc. (Atlanta, GA, USA). They were the core device of the entire sensor network. The sensing arrays are illustrated in Figure 3, which consists of a single-core optical fiber containing six FBGs connected to the two demodulators controlled by a computer. Each FBG weld joint was protected by heat shrinkage tubes with diameters of 2 mm and 1 mm to improve the deformation compatibility between the sensors and soil mass. A total of three FBG horizontal layers were laid in the slope, and each layer was divided into two groups named C1, C2, C3, X1, X2, and X3. The horizontal distance between the two groups was 20 cm with a symmetrical distribution, and the horizontal distance between adjacent gratings in the same layer was 15 cm. Each FBG array was connected to four FBG sensors named 1 to 4 from left to right. Before testing, the initial wavelengths of FBG were recorded, and the frequency of the sensing interrogators was set to 1 Hz, with data recorded every 1 s. When the filling height (including the rock layer) reached 40 cm, 60 cm, and 80 cm, the FBG sensors were placed at the specified position of the slope. The procedure was repeated until the full height of the model slope was obtained.

After the soil embankment reaches the designated height, the optical fiber signal is tested for smoothness to ensure that the signal can be received in a timely manner during the slope excavation process. The static model is left for 24 h to fully couple the soil sample with the optical fiber under the action of self-weight stress. During the excavation test of the model, as shown in Figures 2 and 3, a stepped excavation method is adopted, divided into four steps, with each excavation depth being 20 cm. At the same time, the monitoring equipment collects the strain signals inside the slope in real time during the excavation process. After each excavation step is completed, there is a 1 h pause before proceeding to the next excavation step. When the excavation depth reaches 80 cm, the excavation of the slope is stopped, and the excavation process of the slope is completed. After the completion of the excavation, no obvious cracks were found on the surface of the slope model.

To obtain accurate strain values for the slope, temperature compensation of the fiber was necessary. To accomplish this, three thermometers were utilized to monitor the temperature variations at the top, middle, and bottom of the three layers. These thermometers were connected to a multichannel data acquisition instrument to collect the temperature data automatically.

## 3. Results and Discussions

### 3.1. Numerical Simulation Results

3.1.1. Slope Stability Analysis

Table 2 presents the safety factors calculated by the two limit equilibrium methods, which decreased significantly from 2.69 to 1.20 and 2.49 to 1.09, respectively. The analysis results certify that the factors of safety decrease dramatically, indicating that the stability of the slope has decreased due to stepped excavation. It is important to note that the soil parameters used in the analyses remained consistent throughout the study, while the slope morphology had changed, resulting in the formation of an empty slope face. This change in morphology contributed to the reduction in the stability of the slope. Figure 4 shows the slip circle failure position calculated by the LEM and FEM methods, revealing that the slip surface positions obtained by the two methods are in agreement.

**Table 2.** Safety factors of the slope calculated by two limit equilibrium methods.

| Stage | Factors of Safety $F_s$ | |
|:---:|:---:|:---:|
| | Swedish Slice Method | Bishop's Simplified Method |
| I | 2.69 | 2.49 |
| II | 1.66 | 1.53 |
| III | 1.26 | 1.15 |
| IV | 1.20 | 1.09 |

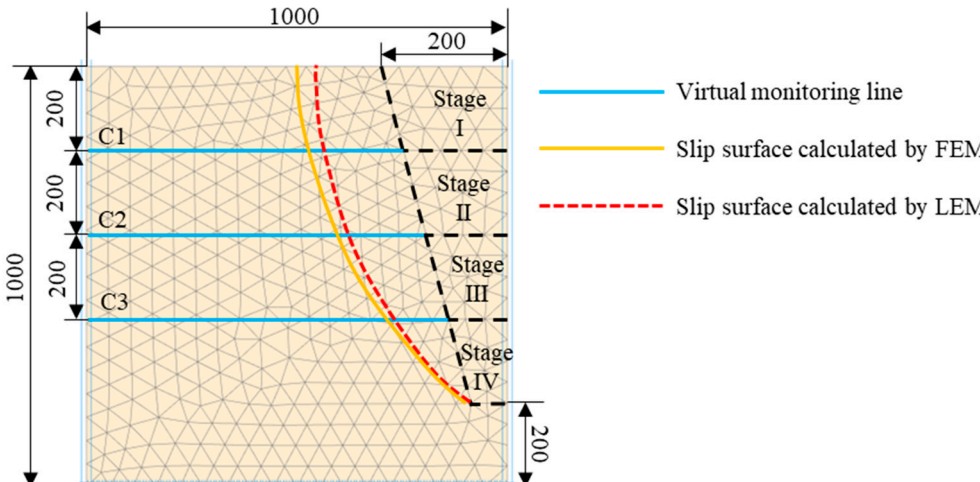

**Figure 4.** Finite element model of excavation and the location of virtual strain monitoring line (unit: mm).

3.1.2. Horizontal Strain Distribution

Through the simulation analysis of the excavation process, the horizontal strain distribution of the specified position of soil mass under stepped excavation was obtained. Three horizontal monitoring lines, labeled layers C1, C2, and C3 from top to bottom, were used. Figure 5 shows the horizontal strain distribution of the three monitoring lines, with distance referring to the left boundary of the slope model. Figure 5 also shows the noted strain in three zones (zone A, B, and C). The strain value gradually increased with each step, and the singular strain area gradually approached the empty face of the slope from C1 to C3.

Figure 6 shows the horizontal strain distribution during the excavation of the four steps. In stage I, the slope mass's strain value was very small, but the foot of the slope mass was subjected to compressive strains, indicating that the surrounding soil was in a compression-shear state, which truly reflected the stress state under excavation. With continued excavation, the strains of each part of the slope increased rapidly, and the horizontal strains near the slip surface were all distributed unevenly. The horizontal strain distributions of other elevations showed a similar pattern. Stage IV showed that the inhomogeneity of the strain distribution in the soil mass was obvious after the completion of the excavation, reflecting the gradual accumulation of shear strain on the slip surface.

It is worth noting that the singular strain area within the slope changes with excavation. The simulation results of this study show that after the excavation is completed, in addition to the singular strain area at the foot of the slope (zone C), the soil body at the upper part of the slope (zone A) also shows some areas of strain concentration. Therefore, during the actual excavation, in addition to closely monitoring the deformation at the foot of the slope, attention should also be paid to monitoring the deformation of the upper soil in the excavation area.

Figure 6 also shows the critical slip surface calculated by the LEM. Theoretically, the critical slip surface in the soil should be consistent with the abnormal strain zone. However, the critical slip surface calculated by the LEM does not always coincide with the maximum horizontal strain zone of the slope. As the safety factor is 1.09 after excavation (according to Bishop's simplified method), the slope is about to slide. Therefore, there is a large plastic zone in the soil mass that cannot accurately identify the slip surface. Furthermore, the above FEM assumes that the slope is an ideal elastic-plastic body, and the nonuniformity and the non-linear constitutive relation of the slope cannot be accurately reflected [31,32]. This may be the reason for the inconsistency between the critical slip surface and the abnormal strain zone.

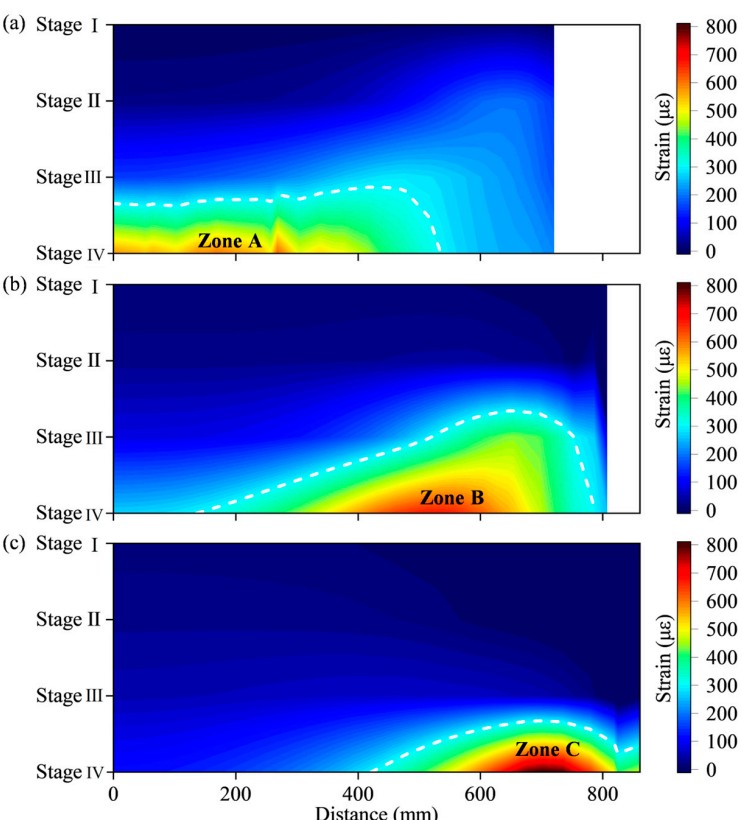

**Figure 5.** Temporal and spatial distribution of horizontal strain from FEM: (**a**) C1; (**b**) C2; (**c**) C3. (Dashed line: a strain of 300 με).

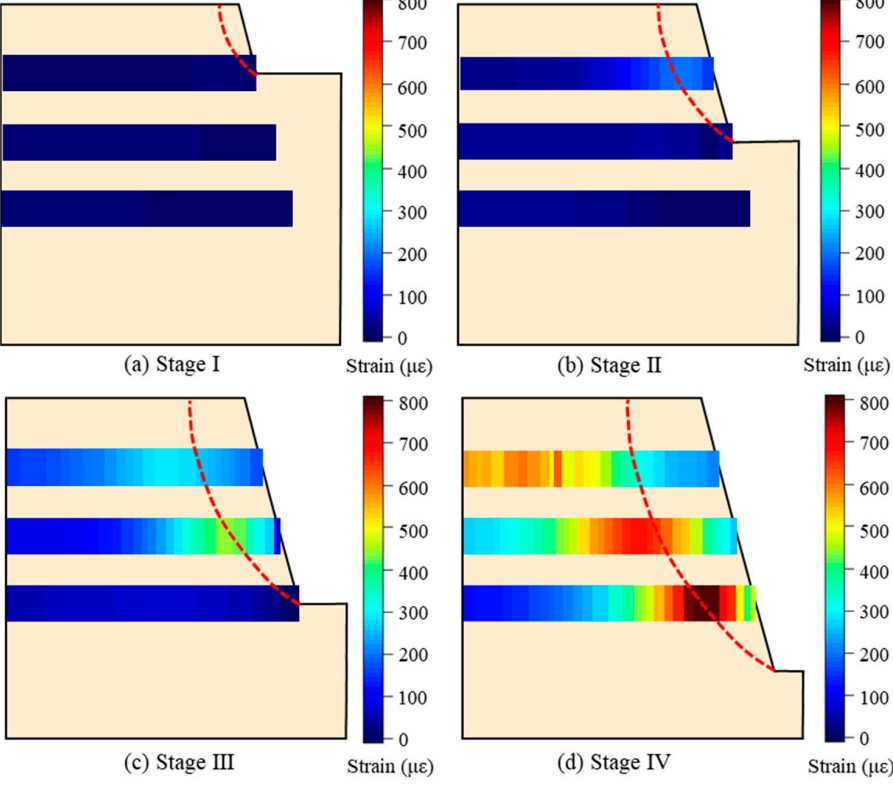

**Figure 6.** Distributions of horizontal strain in monitoring lines under stepped excavation (Dashed line: critical slip surface).

### 3.1.3. Shear Strain Distribution

Figure 7 presents the contour of shear strain in the slope during the excavation process, with the critical slip surface obtained through the strength reduction method (SRM) indicated by the red dashed line. As excavation proceeds, the shear strain at the foot of the slope increases rapidly, and the range of shear strain develops from the foot to the top position, indicating the gradual formation of the critical slip surface. This observation is consistent with the location of the potential sliding surface shown in Figure 6. Therefore, the distribution of the horizontal strains can effectively reflect the stress state within the soil mass. In actual engineering construction, monitoring the strain at the foot of the slope is crucial, as it allows for the timely understanding of strain development in the internal area of the slope.

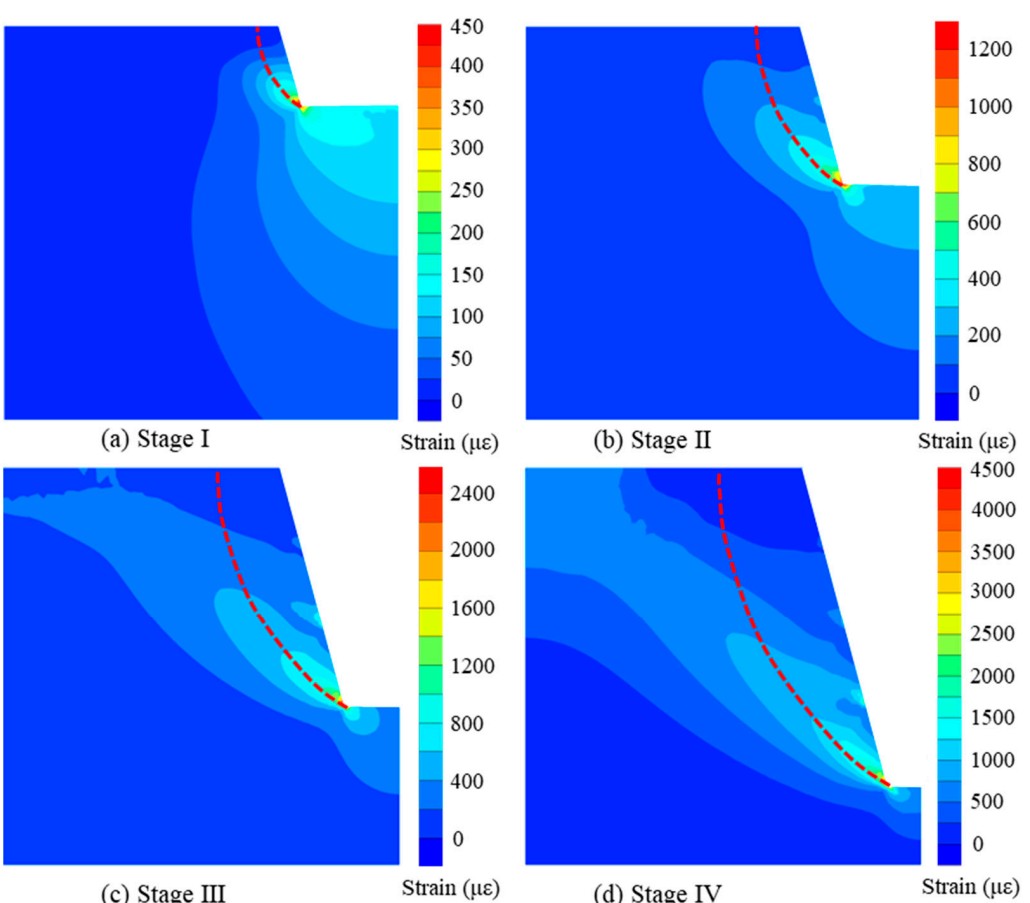

**Figure 7.** Distributions of shear strain in slope mass under stepped excavation: (**a**) Stage I; (**b**) Stage II; (**c**) Stage III; (**d**) Stage IV (Dashed line: critical slip surface).

### 3.2. Physical Model Test Results

3.2.1. Temperature Monitoring

As the experiment was conducted in an underground laboratory, which was almost in a constant temperature environment, the monitoring results from three thermometers indicated that the temperature change was consistent with our expectations, as shown in Figure 8. It was observed that the temperature variation during the test process was within 0.8 °C. Therefore, the necessity to compensate for the effect of temperature on strain measurement can be negligible.

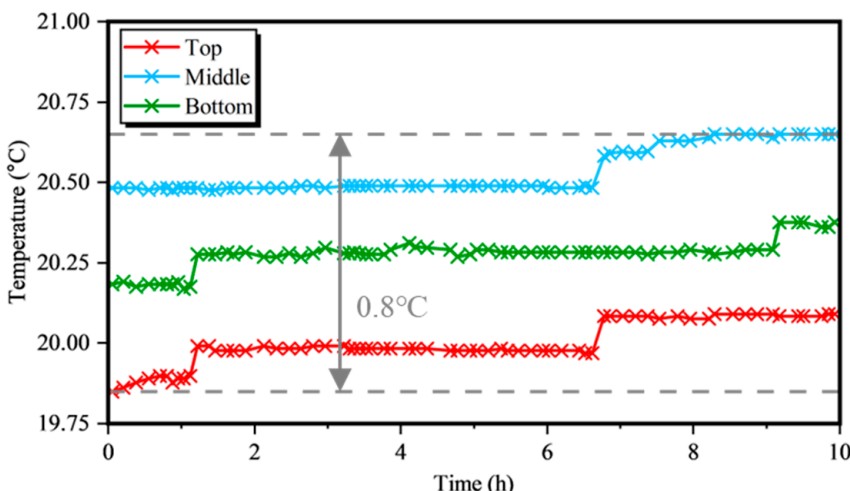

**Figure 8.** Temperature variation measured by the thermometer.

### 3.2.2. Internal Strain of the Soil Mass

Based on the data derived from the fiber demodulators, it was discovered that one of the optical fiber arrays (FBG-X1) with a thin heat shrinkage tube string was damaged. This was likely due to the fibers not being fixed during embedding, which resulted in a lack of deformation coordination between the soil and thin heat shrinkage tubes. Therefore, the wavelength variation data of the remaining five optical fiber arrays were collected. By neglecting the effect of temperature and utilizing Equations (1) and (2), the strain variation information for the specified position of the soil mass was obtained.

Figure 9 depicts the strain distribution of the remaining five optical fiber arrays in the slope. During the excavation, the fiber strain values exhibited a stepped-change trend in general. The monitoring results are outlined as follows:

In stage I, the horizontal strain was negligible, indicating that the soil was not significantly affected by the excavation. However, FBG-C1-4 showed a slight increase in strain, implying a right sliding trend near the slope face, resulting in momentary tensile strain. Similarly, likely due to the proximity of the excavation location, FBG-C2-4 and FBG-X2-4 also experienced sudden strain instability in stage II.

The monitoring data in Figure 9a–e show that the FBG monitoring data differed in each monitoring line, but its trend was consistent, which can reflect the changes in the internal strain field of the slope during the excavation process. From the above observations, it can be concluded that each phase of excavation could get a rapid response to the horizontal strain magnitudes of the optical fiber. From the monitoring results of four FBG sensors on a monitoring line, the strain variations of the optical fiber sensor showed a stepped-increasing tendency in most cases. The strain changes were greater closer to the slope face, with the highest values observed in the middle. Furthermore, the majority of the observed strain was tensile. However, with the excavation of the slope, the strain near the slope surface had a tendency to be gradually transformed into compressive strain (e.g., FBG-C3-4 and FBG-X3-4), which can be clearly seen in Figure 9c,e, indicating that the soil at the foot of the slope near the slope surface was compressed by the upper layer of soil and there was a stress concentration phenomenon.

During the excavation, the largest horizontal strain occurred near the middle of the monitoring line and gradually accumulated, indicating that the critical slip surface of the slope was forming. After the completion of the whole excavation procedure, the strain field of the slope mass tended to be stable, with a brief transition indicating that the soil mass had a very small creep effect. The maximum tensile strain appeared in layer C2 of the second FBG compared with other monitoring lines, up to 900 μ$\varepsilon$, indicating plastic strain accumulation in this location, which was likely situated in the critical slip surface.

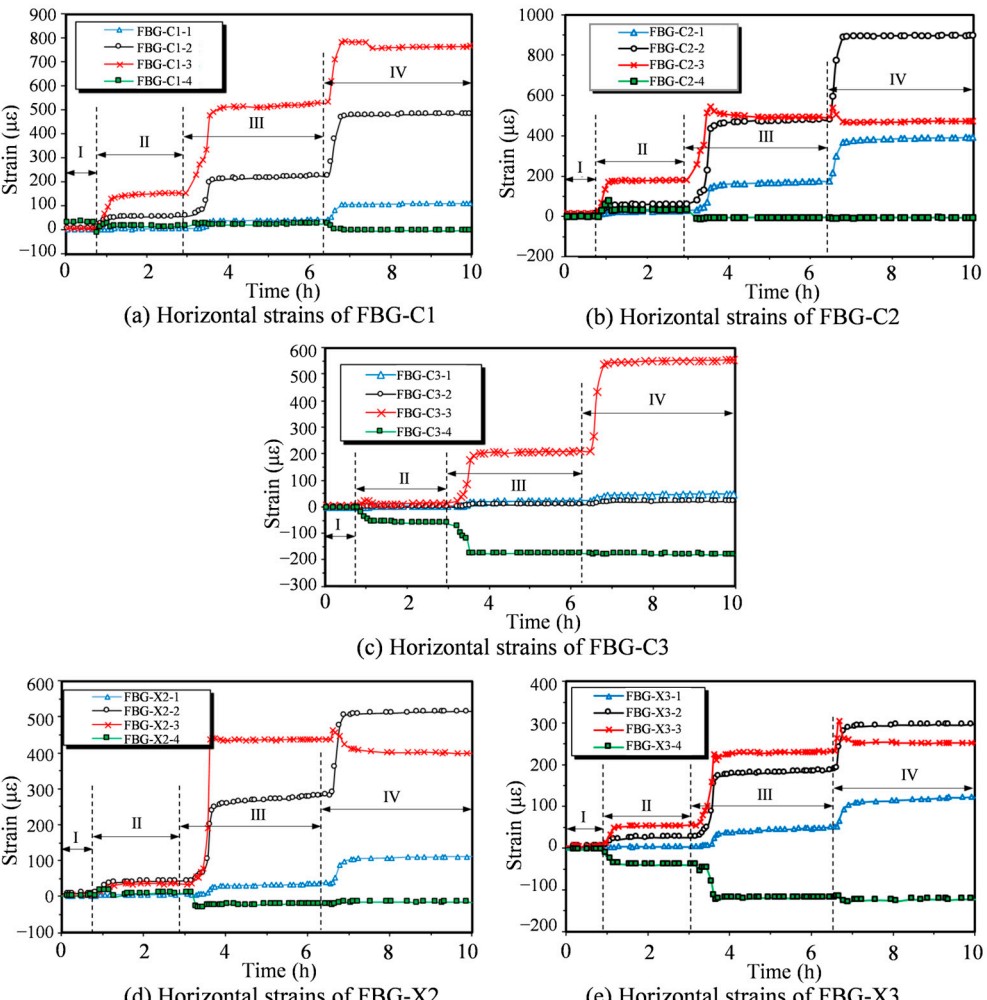

**Figure 9.** Strain monitoring results of the slope model (+: tensile strain, −: compressive strain).

Comparing Figure 9d,e with Figure 9b,c, respectively, the trends were similar, but the strain magnitudes of the FBG sensors with thin heat shrinkage tubes were generally smaller than the thick ones, proving the relatively poor deformation compatibility between the optical fiber and soil mass, and indicating that the deformation of the soil could not be accurately measured.

It is worth noting that some strain points showed drift phenomena, such as FBG-C2-3, which showed strain values that remained unchanged from stage III to IV, and FBG-C3-1 and FBG-C3-2, which showed strain values that remained unchanged from stage I to IV, indicating that these arrays could have been broken due to the low deformation compatibility or artificial disturbance, as shown in Figure 10.

Given the small distance between the sensors and heat shrinkage tubes, the two sides of the heat shrinkage tubes served as anchors in the middle of FBG sensors. The variation in the central wavelength represents the average strain within this spacing, and the strain monitoring value should be considered the average strain of the middle part of the heat shrinkage tube from A to B, as shown in Figure 10a. Therefore, this type of FBG can be viewed as a long-gauge sensor.

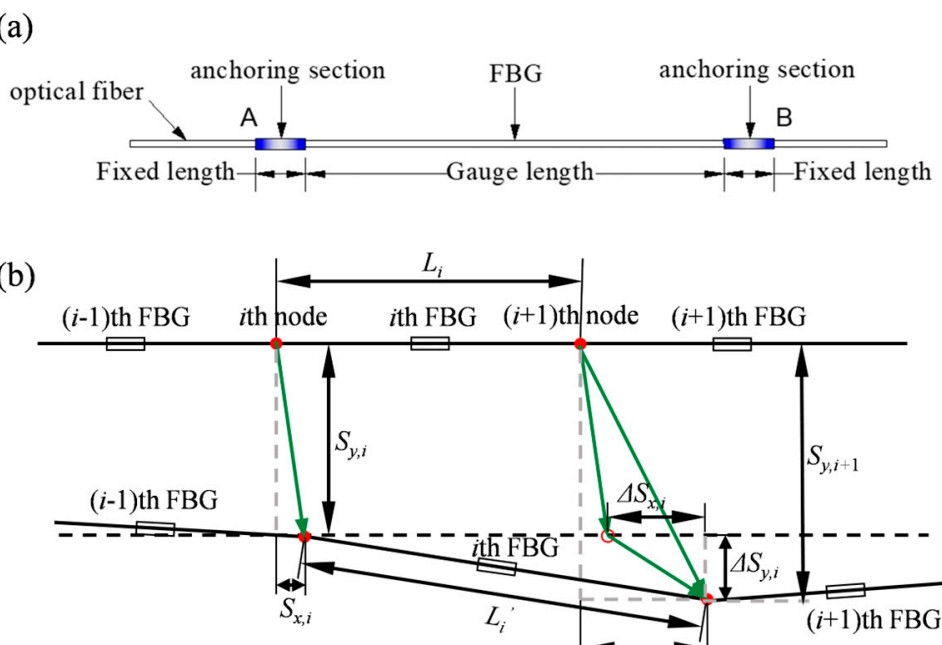

**Figure 10.** (**a**) Structure of the long gauge FBG sensor; (**b**) Deformation mechanics analysis of FBG fiber grating.

Figure 10b reveals the deformation mechanics of the FBG fiber grating subjected to external forces. $S_{y,i}$ and $S_{y,i+1}$ represent the vertical displacement of the *i*-th node and the *i*+1-th node, respectively. When FBGs are subjected to external forces, the distance between two adjacent nodes is increased from $L_i$ to $L_i'$, and the axial strain $\varepsilon_x$ can be expressed by

$$\varepsilon_x = \frac{L_i' - L_i}{L_i}, \tag{5}$$

An algorithm and a MATLAB program were developed and applied to process the data obtained from the FBG sensors and FEM, and the results are plotted in Figure 11. The strain distribution curves from both the FBG sensors and FEM are projected in the slope model, represented by orange and blue lines, respectively. The critical slip surface is illustrated by dashed red lines calculated by the LEM. As shown in Figure 11, the strains of the optical fiber gradually increase during the excavation process, and the strains accumulate adjacent to the points of intersection of the fibers and the critical slip surface overall. The strain peaks of the soil mass appear in the middle area compared to either side. Furthermore, the monitoring results and numerical simulation results are in general agreement, which demonstrates that the intermediate position of the soil mass is within the range of the critical slip surface.

However, there is a noticeable difference between the monitoring results and the FEM results in certain areas. The FEM analysis considers the soil as an elastic continuum medium, which may not be completely objective because, in reality, soil behaves as a porous, elastic-plastic, and viscous medium. Furthermore, the FEM results may be affected by unpredictable errors. Hence, the numerical simulation analysis results only provide a qualitative understanding of the deformation behavior of the slope, and thus the differences observed between the monitoring and simulation results are not surprising.

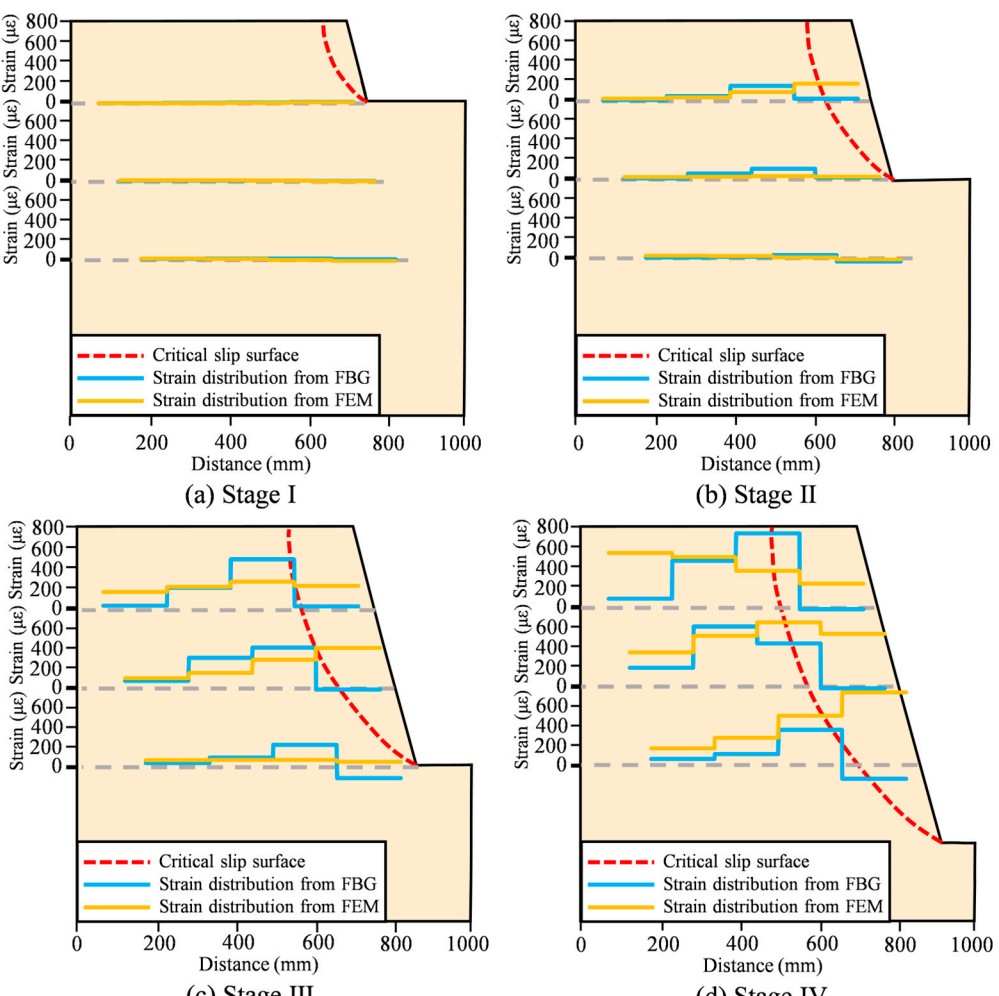

**Figure 11.** Distributions of horizontal strain in slope mass and the critical slip surface under the excavation process (monitoring results versus numerical results).

### 3.3. Relationship between Horizontal Strain and Slope Stability

To investigate the relationship between the slope strains and the corresponding factors of safety, a characteristic strain $\varepsilon_{hc}$ is introduced here, which is defined as the average of maximum horizontal strains from every monitoring section after each step of slope excavation. Table 3 lists the characteristic strains during step-by-step excavation on each monitoring line. Figure 12 presents the relationship curve between the characteristic strain from every monitoring layer and the safety factor after each step of excavation. It was observed that the factor of safety decreased dramatically with an increase in the magnitude of the horizontal strain resulting from slope excavation [33]. The power law provides the best fit to the empirical relationships, i.e.,

$$\varepsilon_{hc} = \frac{\sum\limits_{i=1}^{n} \max(\varepsilon_{hi})}{n}, \tag{6}$$

$$K = a \left( \frac{\varepsilon_0}{\varepsilon_{hc}} \right)^b, \tag{7}$$

where $K$ is the safety factor of the slope, and $a$ and $b$ are the two dimensionless parameters. $\varepsilon_0 = 1 \ \mu\varepsilon$. $\varepsilon_{hi}$ is the measured horizontal strain by the $i$th sensor at the $h$th monitoring layer ($I = 1, 2, \ldots \ldots, \text{n}$).

**Table 3.** Maximum horizontal strain on each monitoring line during slope excavation.

| Stage | Maximum Horizontal Strain on Monitoring line (μ$\varepsilon$) | | | | | Average |
|---|---|---|---|---|---|---|
| | C1 | C2 | C3 | X2 | X3 | |
| 1 | 35.76 | 7.48 | 4.18 | 42.62 | 9.81 | 19.97 |
| 2 | 149.71 | 176.56 | 16.63 | 83.85 | 54.30 | 96.21 |
| 3 | 525.82 | 543.77 | 204.91 | 588.56 | 233.81 | 419.37 |
| 4 | 788.29 | 906.51 | 556.83 | 694.78 | 300.54 | 649.39 |

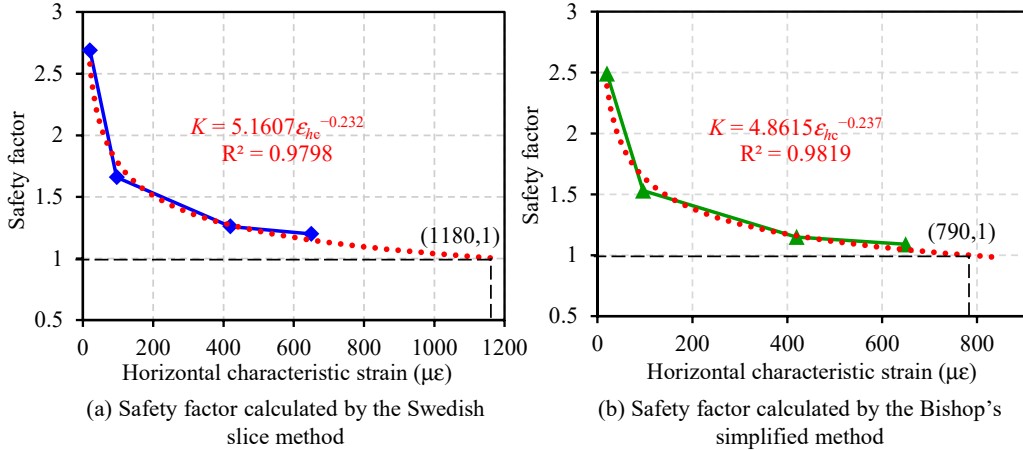

(a) Safety factor calculated by the Swedish slice method

(b) Safety factor calculated by the Bishop's simplified method

**Figure 12.** Relationships between factors of safety and horizontal characteristic strain.

According to the calculation results, there is a good correlation between horizontal characteristic strains and safety factors. The results indicate that it is possible to evaluate slope stability conditions using strain measurements. However, the values of critical strain thresholds corresponding to slope instability are also dependent on the selection of slope safety factor calculation methods. This variation needs to be considered in practical applications. As shown in Figure 12, if the safety factor is set to 1, the corresponding characteristic strains calculated by the Swedish slice method and Bishop's simplified method are 1180 μ$\varepsilon$ and 790 μ$\varepsilon$, respectively. For safety reasons, 790 μ$\varepsilon$ is suggested as the critical characteristic strain threshold for instability prediction for this slope model.

However, the strain is influenced by multiple factors, such as gravel content, soil type, monitoring duration, and others. For instance, the strain thresholds should be carefully determined for complex slopes with gravel or debris inside. In the study by Kogure and Okuda [34], it is recommended that a strain value of 400 μ$\varepsilon$ is sufficient to determine the sliding of a landslide and confirm the location of the sliding surface in the measurement of an actual landslide. Further experimental investigations are required to establish the relationship between strains and safety factors and to derive the critical strain thresholds for stability analysis of complex slopes.

## 4. Conclusions

In this paper, a slope excavation model test was conducted, and FBG sensing technology and numerical simulation were used to measure the soil strain field in a slope model. The feasibility of strain-based slope stability evaluation is explored by analyzing the strain distribution characteristics of the excavation process. The paper analyzed in detail the measured data of slope excavation and the strain distribution characteristics during the excavation process and discussed the relationship between soil strain changes and slope stability. Based on the results obtained, the following conclusions can be drawn:

- The feasibility of using fiber Bragg grating (FBG) sensors to monitor soil strain fields under external forces has been demonstrated. Using the FBG sensor, the changes in horizontal strain in the soil of the slope during the excavation can be monitored.

The monitoring results are in good agreement with the numerical simulation results, indicating that the soil strain information can be effectively monitored using FBG sensing technology;

- Horizontal soil strain during the excavation process can characterize the formation of the critical slip surface. As the excavation proceeds, the horizontal strain in the soil gradually accumulates, with the strain accumulation point being the intersection of the optical fiber and the overall critical sliding surface. Therefore, the strain distribution in the soil provides a clear indication of the formation of the critical sliding surface of the slope;

- The combination of numerical simulations and laboratory tests provides a reference for establishing the relationship between strain parameters and slope stability. By using the finite element method (FEM) and limit equilibrium method (LEM), changes in the internal strain field and safety factor during slope excavation can be simulated. Based on the results of the LEM and the actual strain monitoring data of the laboratory test, the empirical relationship between the strain parameters and the safety factor is established, which is conducive to the real-time evaluation and critical warning of slope stability;

- The coupling between the sensing fiber and the soil significantly influences the test results. Decoupling between the fiber and the soil can occur during excavation if the coupling is poor. The research suggests that using thick heat-shrinkable tubes can provide better interfacial bonding between the soil and the optical fiber compared to thin heat-shrinkable tubes.

It should be noted that fiber optic sensors are more fragile and prone to breakage than traditional sensors. During model construction and excavation, bare optical fiber arrays without protection measures are especially susceptible to breakage. Therefore, it is crucial to install an effective packaging and protection system for these sensors to ensure their proper functioning. Additionally, optimizing the layout of sensing optical fibers is an area that requires further investigation. Further studies are needed to address these issues.

**Author Contributions:** Conceptualization, J.W., H.Z., and W.D.; methodology, W.D.; software, W.Y.; formal analysis, J.W. and W.D.; investigation, W.Y. and W.D.; resources, H.Z.; data curation, H.Z.; writing—original draft preparation, J.W.; writing—review and editing, H.Z. and C.Z. All authors have read and agreed to the published version of the manuscript.

**Funding:** This research was funded by the National Natural Science Foundation of China (Grant No. 42225702) and the National Natural Science Foundation of China (Grant No. 42077235).

**Institutional Review Board Statement:** Not applicable.

**Informed Consent Statement:** Not applicable.

**Data Availability Statement:** The data supporting the reported results can be obtained from the authors.

**Acknowledgments:** The authors thank Zhenyu Wang, Junkuan She, Qian Ma, and Deyang Wang, all from Nanjing University, for their assistance in the experimental works.

**Conflicts of Interest:** The authors declare no conflict of interest.

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
