# Peer review of "Numerical and Experimental Investigation of Slope Deformation under Stepped Excavation Equipped with Fiber Optic Sensors"

_photonics, doi:10.3390/photonics10060692_

Round 1

Reviewer 1 Report

This paper presents a study that involves conducting physical model tests and numerical simulations to evaluate the changes in internal strain and stability conditions during stepped excavation. The authors claim that fiber Bragg grating (FBG) sensors can effectively monitor soil strain information, and that the maximum horizontal strains can be used to assess slope stability conditions. This offers a crucial reference for investigating deformation and failure mechanisms of cut slopes. However, there are several issues that need to be addressed before the paper can be considered for publication. The authors are requested to address the following comments:

(1)  The structure of the Introduction requires further improvement. Instead of merely providing a straightforward description of the literature, the Introduction should present a comprehensive review and summary, including an analysis of the limitations and shortcomings of the current state of this topic. The authors are strongly encouraged to enhance the Introduction section.

(2)  The literature review should be updated to include recent research advances. Some of the papers cited in the Introduction are outdated. The authors are requested to add citations from papers published within the last 2-3 years.

(3)  The authors should introduce essential parameters used in the numerical simulation. In Section 3.3, "Relationship between Horizontal Strain and Slope Stability," the relationship between numerical simulation results and model test results is evaluated. However, it is unclear whether the soil parameters used in these two methods are consistent. Detailed explanations are required. 

(4)  Necessary explanations for annotations in figures must be provided. For instance, on Page 6, Figure 5, the meaning of the white dashed line in the figure should be clarified.

(5)  The novelty of the manuscript needs to be elaborated in detail. For example, the authors should explain why FBG sensors were chosen to monitor the internal strain of the slope.

(6)  The formatting of some images and tables in the text requires revision. On Page 10, Figure 9, a space must be inserted between the name and unit of the horizontal axis; and on Page 13, Table 3, the unit should be provided. The authors are advised to carefully review the manuscript for similar issues.

(7)  It is strongly recommended that the authors seek the assistance of a native English speaker to thoroughly polish the language and expression used in the manuscript.

Author Response

Response to Reviewer 1 Comments

Point:

This paper presents a study that involves conducting physical model tests and numerical simulations to evaluate the changes in internal strain and stability conditions during stepped excavation. The authors claim that fiber Bragg grating (FBG) sensors can effectively monitor soil strain information, and that the maximum horizontal strains can be used to assess slope stability conditions. This offers a crucial reference for investigating deformation and failure mechanisms of cut slopes. However, there are several issues that need to be addressed before the paper can be considered for publication. The authors are requested to address the following comments:

Response:

Thank you for taking the time to review our manuscript and for providing us with your positive feedback. We appreciate the opportunity to improve our work based on your suggestions, and we are pleased to inform you that we have addressed each of your comments as outlined. We have made all modifications in the revised manuscript using the "Track Changes" feature for your convenience during the review process. Once again, we are grateful for your time and helpful feedback, which has helped us to enhance the quality of our manuscript.

Point 1:

The structure of the Introduction requires further improvement. Instead of merely providing a straightforward description of the literature, the Introduction should present a comprehensive review and summary, including an analysis of the limitations and shortcomings of the current state of this topic. The authors are strongly encouraged to enhance the Introduction section.

Response 1:

Thank you for your valuable comment. We appreciate your suggestions and understand the importance of a well-structured introduction that provides a comprehensive review and summary of the topic. We have taken your suggestion and made adjustments to the introduction section by reviewing and summarizing previous literature. We have also pointed out the shortcomings of previous research are the lack of real-time monitoring of strain during slope excavation and the absence of establishing a definite correlation between slope strain and stability.

We have modified the corresponding description in the text and highlighted the changes as follows:

“Slope stability evaluation is a basic problem in geotechnical engineering, particularly in the context of excavation-induced slope failures in urban construction and mining industries. In the past decades, various slope stability analysis methods and techniques have been developed [1]. However, due to the complex nature of slope evolution processes, it remains challenging to understand the deformation and failure mechanism of slope comprehensively and systematically. Excavation is a common cause of slope failures in urban construction and mining [2-4], and it is difficult to evaluate slope stability and predict the field performance in real-time due to the stress state adjustment within the slope during excavation stages and external stress condi-tion changes, which can affect slope deformation and stability [5,6]. Therefore, monitoring the slope’s internal stress state in real-time using geotechnical instrumentation is crucial to make timely assessments of the slope stability and analyze the movement mechanisms , enabling potential failure warnings for the slope.

Geotechnical instruments have been developed and applied to monitor slope be-havior since the 1970s [7]. Traditional monitoring techniques, such as inclinometers, tiltmeters, and extensometers, are easy to operate and have been widely used in ge-otechnical monitoring, but they cannot achieve real-time measurement. Furthermore, many instruments based on resistance, vibration line, or inductive sensor have issues with zero drift because their readings are affected by electrical noise in the environ-ment. In addition, these electrical transducers generally have poor moisture resistance and can hardly achieve long-distance measurement [8,9].

In the past decades, fiber optic sensing technology has been increasingly used to monitor the internal strain and deformation of slopes, proving to be a feasible and ef-fective solution. Compared with other sensors, optical fiber sensors have the ad-vantages of small size, high precision, and immunity to electromagnetic interference. They can also establish a fully or quasi-distributed sensing network combining the wavelength division multiplexing (WDM) and time division multiplexing (TDM) tech-niques to achieve real-time measurement and data transmission [10-12]. Compared to field methods, physical modeling has the advantages of shorter-term testing, lower cost, and more intuitive outcomes [13]. In recent years, several laboratory studies have applied fiber optic sensing technology to slope monitoring. For example, to demon-strate the efficacy of fiber-optic monitoring, Zhu et al. [14], Song et al. [12], and Sun et al. [15] investigated changes in internal strains in slopes using fiber-optic monitoring technology during local loading, slope cutting, and geogrid reinforcement, respectively. The results of these studies confirmed the accuracy of fiber-optic monitoring technol-ogy for measuring internal strains in soil. Since the fiber optic cable can only measure the axial strain distributions, it becomes imperative to establish an analytical model that can effectively characterize the relationship between the strain on the optical fi-bers and the deformation of the soil. Wu et al. [16] proposed a strain integration method to convert strain measurements into shear displacements. Wang et al. [17] proposed a new analytical method for calculating the shear displacement of critical interfaces in landslides, and they demonstrated its feasibility through laboratory shear tests. Xu et al. [18] affixed FBG onto a rubber strip and established the correlation be-tween the bending angle, strain of the rubber strip, and surface deformation of the slope model. To monitor the anchor force of the slope anchor system, a new method was proposed to calculate the slope strain and anchor force considering temperature changes based on optical frequency domain reflectometry (OFDR) technology was proposed [19], and successfully established the relationship between the strain and the anchoring force of the slope anchoring system. While these laboratory experiments have shown that the fiber optic sensing monitoring technology can measure the inter-nal strain of the soil and sense soil deformation [20,21], there have been relatively few attempts to monitor slope strain and its relationship with slope stability during slope excavation, posing a pressing need to address the evolution of slope stability in such scenarios.

Numerous fiber optic monitoring techniques have been developed to monitor the deformation of geotechnical structures. These technologies include fiber Bragg grating (FBG), Brillouin optical time domain reflectometry (BOTDR), Brillouin optical time domain analysis (BOTDA) and so on. Among these technologies, the quasi-distributed FBG is one of the most widely used techniques. FBG sensors can be connected in series or multiplexed to achieve high-resolution measurements, excellent corrosion resistance, and long-term durability. Additionally, FBG sensors can be used for multi-parameter measurements and in sensor networks [9, 22, 23]. Due to the characteristics of FBG sensors, mul-tiple sensors can be connected in series to measure strains at various locations during slope excavation. By combining these measurements with the results of numerical simulation, it is expected to explore the relationship between slope stability and inter-nal strains during excavation.

In this paper, the capability of quasi-distributed fiber optic sensing technology in slope stability monitoring under slope excavation conditions is investigated through physical model tests and numerical simulation methods. The fiber Bragg grating (FBG) sensor was buried in the soil mass to measure the horizontal strain distribution of the slope model under stepped excavation. Combined with the finite element method (FEM) and limit equilibrium method (LEM), the relationship between the slope safety factor and the strain parameter is determined, and the feasibility of strain-based eval-uation of slope stability is discussed. The feasibility of FBG sensing technology in slope stability evaluation and landslide warning is demonstrated.”

Point 2:

The literature review should be updated to include recent research advances. Some of the papers cited in the Introduction are outdated. The authors are requested to add citations from papers published within the last 2-3 years.

Response 2:

Thank you for your comment. We agree that it is essential to include recent research advances in the literature review to provide a comprehensive overview of the current state of the topic. In response to your comment, we have updated the introduction by adding 7 recent papers on the monitoring of strain in slopes using optical fibers. The following is a list of these papers:

[13] Zhang, D.; Wang, J.C.; Zhang, P.S.; Shi, B. Internal strain monitoring for coal mining similarity model based on distributed fiber optical sensing. Measurement. 2017, 97, 234-241. [https://doi.org/10.1016/j.measurement.2016.11.017]

[15] Sun, Y.J.; Cao, S.Q.; Xu, H.Z.; Zhou, X.X. Application of Distributed Fiber Optic Sensing Technique to Monitor Stability of a Geogrid-Reinforced Model Slope. Int. J. of Geosynth. and Ground Eng. 2020, 6, 29. [https://doi.org/10.1007/s40891-020-00209-y]

[16] Wu, H.; Zhu, H.H.; Zhang, C.C.; Zhou, G.Y.; Zhu, B.; Zhang, W.; Azarafza, M. Strain integration-based soil shear displacement measurement using high-resolution strain sensing technology. Measurement. 2020, 166, 15, 108210. [https://doi.org/10.1016/j.measurement.2020.108210]

[17] Wang, D.Y.; Zhu, H.H.; Wang, J.; Sun, Y.J.; Schenato, L.; Pasuto, A.; Shi, B. Characterization of sliding surface deformation and stability evaluation of landslides with fiber–optic strain sensing nerves. Eng. Geol. 2023, 314, 107011. [https://doi.org/10.1016/j.enggeo.2023.107011]

[18] Xu, H.B.; Zheng, X.Y.; Zhao, W.G.; Sun, X.; Li, F.; Du, Y.L.; Liu, B.; Gao, Y. High Precision, Small Size and Flexible FBG Strain Sensor for Slope Model Monitoring. Sensors. 2019, 19, 12, 2716. [https://doi.org/10.3390/s19122716]

[22] Pei, H.F.; Zhang, S.Q.; Borana, L.; Zhao, Y.; Yin, J.H. Slope stability analysis based on real-time displacement measurements. Measurement. 2019, 131, 686-693. [https://doi.org/10.1016/j.measurement.2018.09.019]

[23] Zhang, L.; Zhu, H.H.; Han, H.M.; Shi, B. Fiber optic monitoring of an anti-slide pile in a retrogressive landslide. J. Rock. Mech. Geotech. 2023, 2023.3.17[https://doi.org/10.1016/j.jrmge.2023.02.011]

Point 3:

The authors should introduce essential parameters used in the numerical simulation. In Section 3.3, "Relationship between Horizontal Strain and Slope Stability," the relationship between numerical simulation results and model test results is evaluated. However, it is unclear whether the soil parameters used in these two methods are consistent. Detailed explanations are required.

Response 3:

Thank you for your comment. As you mentioned, it is important to provide a detailed explanation of the soil parameters used in both methods. The parameters used in numerical simulation method should be as consistent as possible with the parameters used in the model test. Table 1 lists the parameters of the soil used in the numerical simulation method. Since the Harding-soil model is used in the FEM model, unlike the Mohr-colomb model, the soil hardening model considers that the Young's modulus is not a fixed value but is related to the stress level of the soil, so there are three Young's modulus here, namely E50, Eoed and Eur. E50 is the secant modulus of triaxial consolidated undrained shear test. Eoed is the tangent modulus of consolidation test. Eur is the unload-reload modulus of triaxial consolidated drained test.

Table 1. Soil parameters used in the numerical simulation

Unit weights

g  (kN/m3)

Young’s modulus E (MPa)

Shear strength

E50

Eoed

Eur

Cohesion c (kPa)

Dilatancy angle ψ (o)

Friction angle φ (o)

13.2

40

40

120

1.6

15.2

30.4

However, the numerical simulation method can only simulate the case of homogeneous soils, while the soil used in the model test is not homogeneous, but a fine sand. The grain size distribution of the test soil are shown in Figure S1. Therefore, the soil used in the numerical simulation method does not exactly match the actual test due to the limitations of the model itself. The limitations of the numerical simulation method do not necessarily indicate its failure. The purpose of our experiment is not to improve the numerical simulation method, but rather to explore the combination of numerical simulation and model testing to establish a relationship between changes in internal strain and slope stability during slope excavation. As shown in Figure 11, although the numerical values of the two methods are different, both methods demonstrate the development of the critical slip surface, and the strain of the fiber gradually increases during excavation, with strain accumulation near the intersection of the fiber and the critical slip surface. Therefore, despite the lack of complete consistency in the model parameters, the numerical simulation results are essentially consistent with the model test results, demonstrating the feasibility of this method.

Fugure S1 Grain size distribution of the test soil

Figure 11. Distributions of horizontal strain in slope mass and the critical slip surface under the excavation process (Monitoring results versus numerical results).

Point 4:

Necessary explanations for annotations in figures must be provided. For instance, on Page 6, Figure 5, the meaning of the white dashed line in the figure should be clarified.

Response 4:

Thank you for your suggestion. We apologize for the lack of clarity in the figure annotations. The white dashed line in the Figure 5 represents a strain of 300 με. The area inside this curve indicates a strain greater than 300 millimeters, which indicates significant deformation. We have revised the figure caption to include this explanation and ensure that all future figures are clearly annotated.

Point 5:

The novelty of the manuscript needs to be elaborated in detail. For example, the authors should explain why FBG sensors were chosen to monitor the internal strain of the slope.

Response 5:

Thank you for your comment. We agree that the novelty of our manuscript needs to be elaborated in more detail. The novelty of our manuscript is the real-time monitoring of internal strain during slope excavation using fiber-optic monitoring technology. By combining these measurements with the results of numerical simulation, we investigate the relationship between internal strain and slope safety factor. This approach provides a more comprehensive understanding of the deformation behavior of the slope and can contribute to the development of more effective slope stability assessment methods. FBG sensors were chosen to monitor the internal strain of the slope due to their high-resolution measurements, excellent corrosion resistance, and long-term durability. Additionally, FBG sensors can be used for multi-parameter measurements and in sensor networks. To clarify this point clearly, we add a paragraph in “Introduction” to make this easier to follow. The corresponding description is added as follows:

“Numerous fiber optic monitoring techniques have been developed to monitor the deformation of geotechnical structures. These technologies include fiber Bragg grating (FBG), Brillouin optical time domain reflectometry (BOTDR), Brillouin optical time domain analysis (BOTDA) and so on. Among these technologies, the quasi-distributed FBG is one of the most widely used techniques. FBG sensors can be connected in series or multiplexed to achieve high-resolution measurements, excellent corrosion resistance, and long-term durability. Additionally, FBG sensors can be used for multi-parameter measurements and in sensor networks [9, 22, 23]. Due to the characteristics of FBG sensors, mul-tiple sensors can be connected in series to measure strains at various locations during slope excavation. By combining these measurements with the results of numerical simulation, it is expected to explore the relationship between slope stability and inter-nal strains during excavation.”

Point 6:

The formatting of some images and tables in the text requires revision. On Page 10, Figure 9, a space must be inserted between the name and unit of the horizontal axis; and on Page 13, Table 3, the unit should be provided. The authors are advised to carefully review the manuscript for similar issues.

Response 6:

Thank you for your comment. We apologize for the errors in the formatting of some images and tables. We have reviewed the manuscript for similar issues and have made the necessary revisions. Regarding Page 10, Figure 9, we have inserted a space between the name and unit of the horizontal axis. On Page 13, Table 3, we have provided the unit as requested. We appreciate your valuable input, and we have ensured that the formatting of all images and tables is now correct and consistent throughout the manuscript.

Point 7:

It is strongly recommended that the authors seek the assistance of a native English speaker to thoroughly polish the language and expression used in the manuscript.

Response 7:

Thank you for your comment. We have taken your recommendation into consideration and have sought the assistance of a native English speaker to improve the language and expression used in the manuscript. With the assistance of a native English speaker, we have made several revisions to the manuscript and have used “Track Changes” feature for your convenience during the review process. We appreciate your comments and suggestions, which have helped us to improve the quality of the manuscript. Thank you again for your valuable input.

Reviewer 2 Report

In this paper, the stability detection performance of quasi-distributed fiber grating sensing technology in slope excavation process is studied by model test and numerical simulation. Combined with limit equilibrium method and finite element method, the slope safety factor based on strain is obtained. The test process is complete and clear, and the results are reliable, but there are still the following problems.

1. Line 112. In tab 1, Why are there three Young 's modulus, like E50 Eoed Eur ? What is the difference between these parameters?

2. Line 144. How is the optical fiber sensor buried ? Is it practical in engineering ?

3. Line 169. How is the temperature compensation? Is there a specific compensation formula ?

4. Some of the pictures in the text are not clear enough, and it is recommended to replace the clear pictures, such as Figure 9 and Figure 10.

5. In the model test, how the slope is excavated is not introduced. Is the monitoring carried out continuously throughout the excavation process ?

6. How to calculate the relationship between horizontal strain and slope safety factor, and how to value the two dimensionless parameters a and b in Formula (4).

7. In Table 3, the maximum horizontal strain of the same horizontal monitoring line during slope excavation is quite different. Is it reliable to take the average value and then calculate ?

8. In the conclusion, is the numerical simulation consistent with the model test results in the first point ? In the previous discussion, the monitoring results seem to be quite different from the FEM results.

Author Response

Response to Reviewer 2 Comments

Point:

In this paper, the stability detection performance of quasi-distributed fiber grating sensing technology in slope excavation process is studied by model test and numerical simulation. Combined with limit equilibrium method and finite element method, the slope safety factor based on strain is obtained. The test process is complete and clear, and the results are reliable, but there are still the following problems.

Response:

We thank you for your time and positive feedback on the evaluation of the manuscript. It is grateful to have the opportunity to improve our manuscript based on your suggestions. Each comment has been addressed as follows. We have made all modifications in the revised manuscript using the "Track Changes" feature for your convenience during the review process.

Point 1:

Line 112. In tab 1, Why are there three Young 's modulus, like E50 Eoed Eur ? What is the difference between these parameters?

Response 1:

Thank you for your comment. Since the Harding-soil model is used in the FEM model, unlike the Mohr-colomb model, the soil hardening model considers that the Young's modulus is not a fixed value but is related to the stress level of the soil, so there are three Young's modulus here, namely E50, Eoed and Eur. E50 is the secant modulus of triaxial consolidated undrained shear test. Eoed is the tangent modulus of consolidation test. Eur is the unload-reload modulus of triaxial consolidated drained test. Due to the incorporation of stress-related model parameters and its ability to consider the impact of stress paths on these parameters, the Harding-soil model offers applicability across a broader spectrum of soil types.

Point 2:

Line 144. How is the optical fiber sensor buried ? Is it practical in engineering ?

Response 2:

In this experiment, we laid three layers of FBG horizontally on the slope, with each layer consisting of 2 single-mode optical fibers connected in series with 4 FBG sensors to monitor strain changes at the FBG location. To ensure deformation coordination between the fiber optic cable and soil, we used heat shrink tubes to protect the FBG solder joints and improve the surface roughness of the sensing optical cable. When burying the optical fiber, we uniformly compacted the soil sample to a thickness of 3-5 cm in the model box. As the filling height (including the rock layer) reached 40 cm, 60 cm, and 80 cm, we buried the sensing optical cable horizontally.

Field geotechnical bodies often experience large deformation and complex geological conditions, requiring long-term monitoring. As a result, the installation of FBG in the field differs somewhat from laboratory tests. Bare fiber grating sensors are inherently fragile, and without proper encapsulation methods, they are susceptible to damage and therefore unsuitable for long-term monitoring. In field monitoring, FBG is typically bonded to structures such as drainage pipes, geogrids, and anti-slip piles to protect against damage caused by the complex geological conditions on site. Additionally, a sensor protection system can be created on site using fiber-reinforced polymer rods, tubes, and other materials that can be implanted in the geotechnical body.

Point 3:

Line 169. How is the temperature compensation? Is there a specific compensation formula ?

Response 3:

Thank you for your comment. The temperature compensation method for FBG sensing systems can be divided into two categories: direct compensation and indirect compensation methods. The direct compensation method involves utilizing a resistance thermometer, ROTDR fiber, or similar devices to capture temperature changes. The strain change due to temperature change is then calculated by the calibration factors. The indirect compensation method involves compensating for strain by using additional FBG sensors placed in the same sensing area. Additional FBG sensors can measure strain changes caused by temperature changes. The corresponding description is added to “2.1.2 Temperature compensation” in the manuscript as follows:

“2.1.2. Temperature compensation

Based on the principle of FBG sensing technology, any alterations in strain and temperature are directly reflected in the period of the index modulation and the effective index of refraction of FBG. Variations in environmental temperature can impact the Bragg wavelength, which in turn affects the measured strain data. Therefore, compensating for temperature is an essential task for any FBG monitoring system. Two commonly used methods for temperature compensation in this area are direct and indirect methods [9].

Standard FBG sensors are unable to simultaneously measure temperature and strain. To compensate for temperature changes, the direct method involves utilizing a resistance thermometer, ROTDR fiber, or similar devices to capture temperature changes. To compensate for temperature changes when measuring strain using FBG sensors, the following equation can be utilized:

,                                      (3)

where ce and cT are the calibration coefficients of strain and temperature.

The indirect compensation method involves compensating for strain by using additional FBG sensors placed in the same sensing area. While the additional sensor is not impacted by mechanical strain-induced soil or rock deformation, it is affected by temperature changes, which are used to achieve temperature compensation by discriminating the wavelength offset. The resulting strain measurements are then corrected, as follows:

,                                     (4)

where DlT B is the shift in wavelength due to temperature change.”

In this manuscript, we utilized the direct temperature compensation method. As shown in Figure 8, the laboratory maintains a nearly constant temperature environment, with only a minor temperature variation of 0.8℃ during the test.. Therefore, the necessity to compensate for the effect of temperature on strain measurement can be negligible.

Figure 8. Temperature variation measured by the thermometer.

Point 4:

Some of the pictures in the text are not clear enough, and it is recommended to replace the clear pictures, such as Figure 9 and Figure 10.

Response 4:

Thank you for bring this to our attention. We appreciate your feedback regarding the clarity of the images in the manuscript, particularly Figure 9 and Figure 10. We have taken your recommendation into consideration and have worked towards replacing these pictures with clearer visions to enhance the readability of the manuscript.

Point 5:

In the model test, how the slope is excavated is not introduced. Is the monitoring carried out continuously throughout the excavation process ?

Response 5:

Thank you for your question. In the model test, the slpoe was excavated using a stepped method, as shown in Figure 2 and Figure 3, with each excavation depth being 20 cm. The monitoring was carried out continuously throughout the excavation process, with the monitoring equipment collecting strain signals inside the slope in real-time. After each excavation step was completed, there was a 1 hour pause before proceeding to the next excavation step. When the excavation depth reached 80 cm, the excavation of the slope was stopped, and the excavation process of the slope was completed. The corresponding description is added to “2.3. Model Material and Instrumentation” in the manuscript as follows:

“After the soil embankment reaches the designated height, the optical fiber signal is tested for smoothness to ensure that the signal can be received in a timely manner during the slope excavation process. The static model is left for 24 hours to fully couple the soil sample with the optical fiber under the action of self-weight stress. During the excavation test of the model, as shown in Figure 2 and Figure 3, a stepped excavation method is adopted, which is divided into four steps, with each excavation depth being 20 cm. At the same time, the monitoring equipment collects the strain signals inside the slope in real-time during the excavation process. After each excavation step is completed, there is a 1 hour pause before proceeding to the next excavation step. When the excavation depth reaches 80 cm, the excavation of the slope is stopped, and the excavation process of the slope is completed.”

Point 6:

How to calculate the relationship between horizontal strain and slope safety factor, and how to value the two dimensionless parameters a and b in Formula (4).

Response 6:

In this study, the relationship between horizontal strain and slope safety factor was calculated by fitting the experimental data to a power-law function, based on the empirical formula presented in the literature [33] (Equation 4 in the manuscript). The two dimensionless paremeters, a and b, in Equation (4) were then determined from the fitted curves.In this test, different methods of calculating safety factors produced varying results, resulting in different fitted functions and dimensionless parameters a and b. Specifically, in the Swedish slice method, a=5.1607 and b=0.232, while in the results calculated by Bishop's simplified method, a=4.8615 and b=0.237. For specific applications in engineering, laboratory tests or in-situ tests are necessary to determine the values of these parameters based on the actual soil conditions and compaction.

Our proposed method aims to establish a relationship between internal strain and slope stability during slope excavation. However, this formula cannot be applied to all cases without definite parameters, and this assessment method may not be accepted by most people. Therefore, the parameters a and b, as well as the slope stability assessment method, should be adjusted according to the specific situation. When performing the calculation of slope stability, we recommend using the more conservative calculation method of safety factor to ensure the safety of construction."

Point 7:

In Table 3, the maximum horizontal strain of the same horizontal monitoring line during slope excavation is quite different. Is it reliable to take the average value and then calculate ?

Response 7:

Thank you for your comment. Given the high resolution of FBG sensor measurement and its wide application in engineering, we chose FBG sensor to measure the internal strain of the slope in this test. However, due to the quasi-distributed nature of the FBG sensor and its ability to measure the strain at only one point where the FBG is located, it is challenging to detect the strain changes between two FBG sensors.

To enhance the coupling between the soil and the fiber, we used heat-shrinkable tubes to increase the roughness of the fiber surface. As depicted in Figure 10, the small distance between the sensor and the heat shrink tube caused the two sides of the heat shrink tube to act as anchors in the middle of the FBG sensor. Therefore, the change in the center wavelength at the point where the FBG is located also indicates the average strain within that spacing, and the strain monitoring value should be considered as the average strain from A to B in the middle of the heat shrink tube. Thus, this type of FBG sensor can be regarded as a long-dimensional sensor, and the adopted method of calculating the average strain is reliable.

The large variation of strain values measured on the same monitoring line is a realistic and common outcome because, during slope excavation, the area with a higher strain inside the slope will gradually develop into a slip crack zone and cause slope instability. This is one of the primary objectives of this test, which is to observe the process of internal strain variation during slope excavation

Point 8:

In the conclusion, is the numerical simulation consistent with the model test results in the first point ? In the previous discussion, the monitoring results seem to be quite different from the FEM results.

Response 8:

Thank you for your comment. In the conclusion, we noted that the numerical simulation results were generally consistent with the model test results regarding the deformation behavior of the slopes. However, we also observed significant differences between the monitoring results and the finite element results in some areas. This discrepancy arises due to two reasons.

Firstly, the FEM analysis assumes that the soil is an elastic continuous medium, which may not be entirely accurate because the soil is actually a porous, elasto-plastic, and cohesive medium. Thus, it is challenging to maintain consistency between the intrinsic model and soil parameters we choose and the test soil during numerical simulation.

Secondly, errors in the results may arise due to differences in soil compaction size, excavation time, and excavation location during the test process. Therefore, although inconsistency between finite element results and monitoring results is normal.

The purpose of the finite element simulation is not to achieve exact agreement between simulation results and actual monitoring results, but to provide a qualitative understanding of slope deformation behavior. However, the monitoring results are relatively reliable and can accurately measure the internal strain of the slope. In establishing the relationship between slope strain and slope stability, we also used the monitoring results from laboratory experiments.

Reviewer 3 Report

The authors report that the experimental results using an isotropic slope model of 1m long, 0.5m wide, and 1m high are similar to the FEM and LEM pretend results, so the stress concentration at the base of the slope can be monitored from the signals collected by installing the FBG array. I completely agree with the limited experimental conditions.

However, it needs to be demonstrated by further experiments simulating non-uniform soil containing gravel, rocks, etc. in the soil, not in the basic structure shown in Figure 2. If this is supplemented, I think it's worth publishing.

Author Response

Response to Reviewer 3 Comments

Point1:

The authors report that the experimental results using an isotropic slope model of 1m long, 0.5m wide, and 1m high are similar to the FEM and LEM pretend results, so the stress concentration at the base of the slope can be monitored from the signals collected by installing the FBG array. I completely agree with the limited experimental conditions.

However, it needs to be demonstrated by further experiments simulating non-uniform soil containing gravel, rocks, etc. in the soil, not in the basic structure shown in Figure 2. If this is supplemented, I think it's worth publishing.

Response1:

Thank you for taking the time to review our manuscript. Yes, we acknowledge that the experimental conditions in our study are limited, and we used an isotropic slope model to simplify the problem. We did not consider non-homogeneous soils, such as those containing debris and rocks, due to space limitations in the paper.

However, we demonstrated the feasibility of using FBG sensors to monitor internal strains and stress concentrations in slopes during excavation and explored the use of strains to assess slope stability. Your suggestion of considering non-homogeneous soils in the slope model is meaningful and will be the next step in our research.

Our future research will investigate the internal strain variation during excavation for slopes with different gravel contents, such as 5%, 10%, 15%, and 20%, with a uniform grain size of 5mm. We also plan to study the impact of different grain sizes, such as 5mm, 10mm, 15mm, and 20mm, with a uniform gravel content of 20%. This research can also explore the application of FBG sensors in various slope models to improve the understanding of slope deformation behavior and develop more accurate assessment methods for slope stability.

Round 2

Reviewer 3 Report

Dear authors,

As you know, It is a known fact that the safety of a slope can be diagnosed with an FBG sensor, and in actual application, since the environmental conditions are slightly different, even if a sensor with the same specifications is used, the result will be different.

It would be very meaningful if the internal strain and stress concentration of the slope could be monitored by deriving the critical value and data trend that can be judged as abnormal during excavation regardless of the gravel content.

Sincerely

Author Response

Point:

As you know, It is a known fact that the safety of a slope can be diagnosed with an FBG sensor, and in actual application, since the environmental conditions are slightly different, even if a sensor with the same specifications is used, the result will be different.

It would be very meaningful if the internal strain and stress concentration of the slope could be monitored by deriving the critical value and data trend that can be judged as abnormal during excavation regardless of the gravel content.

Response:

Thank you for your comment. In fact, the magnitude of the monitored strain values and the area of strain concentration will be different even if the same sensor is used due to the difference in excavation area, excavation time, soil properties and environmental conditions, and we strongly agree with you on this point. However, we believe that such differences are normal and expected.

In our study, for a simple homogeneous soil slope model, the critical strain thresholds for slope instability derived using different slope safety factor calculation methods are different. As shown in Figure 12, the relationship between the derived safety factor and strain is represented by different functions due to the different safety factor calculation methods used. When the safety factor is 1, the strains calculated by the Swedish slice method and the Bishop’s simplified method are 1180 me and 790 me, respectively. To be on the safe side, for our homogeneous soil slope model, the critical strain threshold for slope instability is 790 me.

 Regarding laboratory tests of complex slope models containing gravels, as we mentioned in our first reply, this will be our next step. We regret that further tests are needed because the slope model with gravels is more complex. For example, Kogure and Okuda [34] argue that 400 me is sufficient to determine the sliding of a landslide and thus confirm the location of the slip surface in the measurement of an actual landslide. This is because the strain values are related to gravel content, soil type, duration of monitoring and other factors. Thus, a reliable relationship between strain and slope stability of a complex slope model cannot be derived by derivation alone. We have added the relevant expressions in “3.3. Relationship between Horizontal Strain and Slope Stability” in the manuscript.

“According to the calculation results, there is a good correlation between horizontal characteristic strains and safety factors. The results indicate that it is possible to evaluate slope stability conditions using strain measurements. But the values of critical strain thresholds corresponding to slope instability is also dependent on the selection of slope safety factor calculation methods. This variation needs to be considered in practical applications. As shown in Figure 12, if the safety factor is set to 1, the corresponding characteristic strains calculated by the Swedish slice method and the Bishop’s simplified method are 1180 me and 790 me, respectively. For safety reasons, 790 me is suggested as the critical characteristic strain threshold for instability prediction for this slope model.

However, the strain is influenced by multiple factors, such as gravel content, soil type, monitoring duration, and others. For instance, the strain thresholds should be carefully determined for complex slopes with gravel or debris inside. In the study of Kogure and Okuda [34], it is recommended that a strain value of 400 me is sufficient to determine the sliding of a landslide and confirm the location of the sliding surface in the measurement of an actual landslide. Further experimental investigations are required to establish the relationship between strains and safety factors, and to derive the critical strain thresholds for stability analysis of complex slopes.”

[34] Kogure, T.; Okuda, Y. Monitoring the vertical distribution of rainfall-induced strain changes in a landslide measured by dis-tributed fiber optic sensing with Rayleigh backscattering. Geophys. Rese. Lett. 2018, 45, 4033-4040. [https://doi.org/10.1029/2018GL077607]
